An optimized ensemble model with advanced feature selection for network intrusion detection

Ahmed Afaq 1
http://orcid.org/0000-0002-6423-9809 Asim Muhammad 2 masim@psu.edu.sa
Ullah Irshad 1
Zainulabidin 3
Ateya Abdelhamied A. 2 4
1 School of Computer Science and Engineering, Central South University , Changsha, Hunan , China
2 EIAS Data Science Lab, College of Computer and Information Sciences, Prince Sultan University , Riyadh , Saudi Arabia
3 Institute of Business and Management Sciences (IBMS), The University of Agriculture , Peshawar, Khyber Pakhtunkhwa , Pakistan
4 Department of Electronics and Communications Engineering, Zagazig University , Zagazig , Egypt
Ardagna Claudio
Electronic publication date: 2024 Nov 26
Publication date: 2024
Volume: 10
Electronic Location ID: e2472
Received 2024 Jun 27; Accepted 2024 Oct 11
Copyright: © 2024 Ahmed et al.
Copyright year: 2024
Copyright holder: Ahmed et al.
License: This is an open access article distributed under the terms of the Creative Commons Attribution License, which permits unrestricted use, distribution, reproduction and adaptation in any medium and for any purpose provided that it is properly attributed. For attribution, the original author(s), title, publication source (PeerJ Computer Science) and either DOI or URL of the article must be cited.
License URL: https://creativecommons.org/licenses/by/4.0/

Keywords: Network intrusion detection systems, Machine learning, Ensemble models, Cybersecurity, Feature selection

Funding: Prince Sultan University This work was supported by Prince Sultan University. The funders had no role in study design, data collection and analysis, decision to publish, or preparation of the manuscript.

==============================
In today’s digital era, advancements in technology have led to unparalleled levels of connectivity, but have also brought forth a new wave of cyber threats. Network Intrusion Detection Systems (NIDS) are crucial for ensuring the security and integrity of networked systems by identifying and mitigating unauthorized access and malicious activities. Traditional machine learning techniques have been extensively employed for this purpose due to their high accuracy and low false alarm rates. However, these methods often fall short in detecting sophisticated and evolving threats, particularly those involving subtle variations or mutations of known attack patterns. To address this challenge, our study presents the “Optimized Random Forest (Opt-Forest),” an innovative ensemble model that combines decision forest approaches with genetic algorithms (GAs) for enhanced intrusion detection. The genetic algorithms based decision forest construction offers notable benefits by traversing a wider exploration space and mitigating the risk of becoming stuck in local optima, resulting in the discovery of more accurate and compact decision trees. Leveraging advanced feature selection techniques, including Best-First Search, Particle Swarm Optimization (PSO), Evolutionary Search, and Genetic Search (GS), along with contemporary dataset, this research aims to enhance the adaptability and resilience of NIDS against modern cyber threats. We conducted a comprehensive evaluation of the proposed approach against several well-known machine learning models, including AdaBoostM1 (AbM1), K-nearest neighbor (KNN), J48-Decision Tree (J48), multilayer perceptron (MLP), stochastic gradient descent (SGD), naïve Bayes (NB), and logistic model tree (LMT). The comparative analysis demonstrates the effectiveness and superiority of our method across various performance metrics, highlighting its potential to significantly enhance the capabilities of network intrusion detection systems.

Introduction

Network Intrusion Detection Systems (NIDS) are essential components of cybersecurity, tasked with monitoring network traffic to swiftly detect unauthorized activities. These systems are crucial for defending networks against a variety of threats, including viruses, hackers, and insider attacks, thereby ensuring the availability, confidentiality, and integrity of data and resources within business networks (Agarwal & Das, 2023; Anisetti et al., 2023). As network security threats continue to evolve, it is imperative to advance the development and deployment of NIDS to protect digital assets from increasingly sophisticated intrusions. As the digital landscape evolves and cyber threats become more sophisticated and prevalent, the significance of NIDS cannot be overstated. With each passing day, new vulnerabilities emerge, and cybercriminals devise increasingly ingenious methods to exploit them. Therefore, it is imperative to not only enhance the existing NIDS capabilities but also to develop and deploy innovative solutions that can effectively counter these evolving threats. By staying ahead of the curve and embracing cutting-edge technologies and strategies, organizations can fortify their defenses and safeguard their digital assets against the ever-present specter of cyber intrusion. Collaborating with industry peers and participating in information-sharing initiatives further enhances cybersecurity.

Traditional machine learning (ML) techniques have been extensively utilized in NIDS for their high accuracy and low false alarm rates. However, these techniques often fall short in detecting innovative and complex threats, particularly mutation attacks that involve subtle modifications of known attack patterns to evade detection. Intrusion detection systems have been implemented in Khaliq (2020) and Mohammadpour et al. (2022) by using two separate methodologies; (a) Anomaly-based detection entails continual study of crucial network features and network traffic monitoring. It monitors and analyses network traffic, sending out alerts in the event that it finds unusual or atypical activity. However, (b) signature-based detection depends on the preservation of established attack patterns, or “signature.” The system looks for certain patterns in network packets, and when a match is discovered, it alerts the user to the harmful activity. The examination of network behaviour characteristics forms the basis of anomaly-based identification systems. By closely examining massive amounts of data, seeing spikes in traffic to or from a particular host, and spotting imbalances in network load, this kind of detection can spot unusual activity. One problem with this type of method is that if the malicious behaviour is consistent with regular network behaviour (Ullah et al., 2023), it will not be identified as an abnormality. A major advantage over signature-based detection is that a novel attack for which no signature exists can be identified if it acts differently from usual traffic behaviour patterns.

Despite the availability of numerous attack detection systems, many are not highly effective at detecting and analyzing intrusions or malicious activities. Typically, anomaly-based detection systems are developed by integrating various machine learning approaches to predict network breaches. Previous studies have commonly utilized datasets such as KDD-Cup99 (Kavitha, Uma Maheswari & Venkatesh, 2021), KDD98 (Almseidin, Al-Sawwa & Alkasassbeh, 2022), and NSL-KDD7 (Ahmed, Hameed & Bawany, 2022). However, with the rapid advancements in internet technology and the emergence of new threats, it has become imperative to prioritize the use of more recent datasets to ensure the relevance and effectiveness of intrusion detection systems. To enhance the functionality of network intrusion detection systems, it is crucial to employ updated datasets that reflect contemporary network activities and attacks. Modern datasets, such as UNSW-NB15, provide a more accurate and efficient basis for evaluating network intrusion detection systems. This study constructs a framework for network attack detection using the UNSW-NB15 dataset, which includes recent network attacks and normal activity (Belhadj aissa, Guerroumi & Derhab, 2020; Louk & Tama, 2023). Traditional machine learning techniques have been widely used for network intrusion detection due to their high accuracy and low false alarm rates. However, these methods often struggle with detecting sophisticated and evolving threats, particularly those involving subtle variations or mutations of known attack patterns. To effectively address these challenges and improve the detection of novel and complex intrusions, it is crucial to develop more robust and intelligent approaches. Traditionally, NIDS benchmarking has relied on outdated datasets such as KDD-Cup99, KDD98, and NSL-KDD, which no longer reflect the current threat landscape. Leveraging the recent UNSW-NB15 dataset addresses these limitations and offers a more accurate and comprehensive basis for developing advanced network intrusion detection systems (Zohaib, Asim & ELAffendi, 2024).

To address the limitations of traditional machine learning techniques in network intrusion detection, we introduces the “Optimized Random Forest (Opt-Forest),” an innovative ensemble model for constructing decision forests. This model leverages advanced optimization techniques, including Best-First Search, Particle Swarm Optimization (PSO), Evolutionary Search, and Genetic Search (GS). The hybrid approach enhances the accuracy, robustness, and efficiency of the decision forest, outperforming traditional optimization algorithms. The genetic algorithms based decision forest approaches offers several advantages, such as exploring a broader search space and reducing the likelihood of getting trapped in local optima. This results in more accurate and compact decision trees. Additionally, the flexibility of the GAs framework allows for optimizing multiple objectives, such as classification accuracy and tree size. The proposed approach also is evaluated against well-known machine learning models, including AdaBoostM1 (AbM1), K-nearest neighbor (KNN), J48-Decision Tree (J48), multilayer perceptron (MLP), stochastic gradient descent (SDG), naïve Bayes (NB), and logistic model tree (LMT). The comparative analysis demonstrates the effectiveness and superiority of our method across various performance metrics.

The article’s primary contributions are as follows: Introduction of Opt-Forest model: This study introduces the Optimized Random Forest (Opt-Forest), a novel ensemble model combining decision forest approaches with genetic algorithms (GAs). This integration enhances the detection of sophisticated and evolving cyber threats, surpassing the capabilities of traditional machine learning methods.

Advanced feature selection techniques: Leveraging advanced feature selection techniques such as Best-First Search, PSO, Evolutionary Search, and GS, the study ensures the selection of the most relevant features. This contributes to the model’s ability to maintain high accuracy and low false alarm rates.

Comprehensive evaluation: The research includes a thorough evaluation of the Opt-Forest model against several established machine learning algorithms, including including AbM1, KNN, J48, MLP, SDG, NB, and LMT. This comparative analysis demonstrates the superior performance of Opt-Forest across various performance metrics.

Use of contemporary dataset: By utilizing the most current UNSW-NB15 dataset, the study underscores the importance of using modern, real-world data to enhance the precision and effectiveness of intrusion detection systems. This approach ensures that the findings are relevant and applicable to current cyber threat landscapes.

The article has been divided into the following sections: “Related Work” explores the relevant literature and provides an overview of earlier studies. “Preliminaries” covers the preliminaries, outlining foundational concepts necessary for understanding the proposed approach. “Proposed Methodology” details the methodology and study design process. The results, along with a thorough analysis, are presented in “Experimental Results and Analysis”. Finally, “Conclusion” draws conclusions from the study, summarizing key findings and discussing their implications.

Related work

The increasing prevalence of sophisticated cyber threats necessitates the development of advanced NIDS to protect sensitive digital assets. With the advent and emergence of new methodologies, computer networks use the most up-to-date technologies to implement them (Alrayes et al., 2023), which has radically modified the degree of threats (Tama & Lim, 2021). Thus, the dataset known as UNSW-NB15 was developed to target current network threat categories. In Fathima et al. (2023), Experts constructed a model based on the categorization of attack groups seen in the UNSW-NB15 dataset. The study employed the Association Rule Mining approach to select features. The Expectation-Maximization (EM) approach and naïve Bayes (NB) algorithm (Alshammri et al., 2022) were used for classification. But neither system’s performance in identifying rare attacks showed much improvement; the Expectation-Maximization method produced 58.88% accuracy, while naïve Bayes approach got 78.06% accuracy. In Kumar et al. (2020), an unified calcification-based NIDS was unveiled, including information gain (IG) for feature selection, a decision tree, and a combination of clusters formed employing the K-means approach. The NIT Patna CSE lab (RTNITP18) dataset was used for a test dataset to assess the proposed methodology, with the research concentrating on 22 attributes and four categories of network intrusion from the UNSW-NB15 dataset. The accuracy of the proposed model was 84.83%, whereas the accuracy of the DT C5 model was 90.74%.

Recent studies Lee, Pak & Lee (2020) have applied deep learning to NIDSs to enhance classification accuracy. To address the challenges of high-dimensional data and slow detection, one approach uses a deep sparse autoencoder for feature extraction, followed by classification with a random forest (RF) algorithm. This method improves detection speed and accuracy, achieving 99% accuracy in distinguishing normal from attack traffic. However, further research is needed to improve performance for sparse classes. In Kasongo & Sun (2020), a complex network intrusion detection system was proposed, combining the feature selection method of XGBoost algorithm’s with a total of five classification algorithms (LR, KNN, ANN, DT, and SVM). The study applied multiclass and binary classification approaches to the UNSW-NB15 dataset. With the KNN classifier, multiclass classification had a lesser accuracy of 82.66%, while binary classification did well with a 96.76% accuracy. In Kumar, Das & Sinha (2021), a concept for a Unified Intrusion Detection System (UIDS) that can distinguish between legitimate flow and four various kind of network assaults resulted using of the UNSW-NB15 dataset. The proposed UIDS model was developed by combining rules (R) from several DT models, including IG for feature selection and K-means clustering. Using methods like support vector machines, neural networks, and C5 so, this model had to trained. As an outcome, the model that was suggested outperformed the current strategies, with 88.92% accuracy. As opposed to this, the accuracy of other algorithms, such as SVM, neural network, and C5, was 78.77%, 86.7%, and 89.76%, respectively.

The efficiency of both ML and data mining techniques in spotting intrusions in the IoT system is described in a survey study (Saheed, 2022) by running the algorithms in IDSs and recognizing abnormalities or categorizing the traffic. In Ajdani & Ghaffary (2021), SVM is used for detecting intrusions. They also employed the feature elimination approach to boost efficiency. Using the suggested feature reduction strategy, they picked the top nineteen features from the KDD-Cup99 dataset. The suggested approach employs a relatively tiny dataset. To detect network threats, a two-step anomaly based NIDS approach was utilized in Kao et al. (2022). The suggested technique combined RF feature selection algorithms with logistic regression (LR), recursive feature elimination (RFE), gradient boost machine (GBM), and SVM in the framework of a comprehensive analysis of the UNSW-NB15 dataset. The outcomes showed how the multi-classifiers employing DT had an accuracy about 86.04%. Researchers used KDD-Cup99 and UNSW-NB15 in their work (Choudhary & Kesswani, 2020) to use a genetic algorithm (GA) by merging with the logistic regression wrapper-based feature selection method. Out of 42 features utilizing 20 features in the UNSW-NB15 feature space, the GA-LR merged with the decision tree classifier an accuracy of 81.42% and a false alarm rate of 6.39% across multiple simulations. Furthermore, the GA-LR and the DT classifier with 18 features on the KDD-Cup99 dataset have an accuracy of 99.90% and a false alarm rate of 0.105%. In this study Injadat et al. (2020), efforts have been made to develop ML-based Network Intrusion Detection Systems (NIDSs) that achieve a balance between computational efficiency and detection performance. A multi-stage optimized ML framework is introduced to reduce training sample sizes and enhance feature selection. The work also investigates hyper-parameter optimization techniques, achieving over 99% detection accuracy with the CICIDS 2017 and UNSW-NB 2015 datasets, surpassing existing models in both accuracy and false alarm rates.

In Dickson & Thomas (2021), after reducing characteristics in the UNSW-NB15 dataset using a random forest technique, researchers were able to identify 11 important attributes. They investigated machine learning methods for classification, testing F-measure and accuracy using test data, including KNN, decision tree, Bagging Meta Estimator, and RF. One-hot encoding was used to convert precise characteristics in the UNSW-NB15 dataset. After several testing, the newly presented information gain two-stage approach obtained an 85.78% accuracy and 15.78% false alarm rate. An NIDS architecture was developed in Kanimozhi & Jacob (2019) by researchers using Synthetic Minority over Sampling (SMOS) for rise in minority cases and One-Side Selection (O-SS) to decrease noisy data records in majority classes. Bidirectional long short term memory (Bi-LSTM) methods picked temporal information, whereas convolutional neural networks (CNN) retrieved spatial variables. The suggested deep learning (DL) model (Hussain et al., 2024), which is a combination of CNN and Bi-LSTM, was assessed using accuracy as the main performance indicator on the UNSW-NB15 and NSL-KDD datasets. An intrusion detection system based on an advanced principal component (APCA) algorithm and an incremental extreme learning machine (IELM) approach was developed in Kumar et al. (2022). Key characteristics for the best attack prediction by IELM were found via APCA. Using the UNSW-NB15 dataset, the researchers assessed the IDS with an emphasis on accuracy, detection rate (DR), and false alarm rate (FAR). Based on testing data, the IELM-APCA obtained a 70.51% accuracy rate, a 77.36% DR, and a 35.09% FAR.

This study’s suggested method for network intrusion detection uses a “Opt-Forest” ensemble model based on decision forest and GAs. Unlike previous studies in the literature, which primarily focused on individual machine learning algorithms, this proposed system combines the power of a decision forest with the optimization capability of a genetic algorithm for the enhancement of the accuracy and efficiency of NID. The “Opt-Forest” method comprises creating a decision forest and using a GAs to select the optimal sub-forest from it. The GAs is starting population is made up of the finest trees. This novel strategy seeks to enhance the performance of NID via optimizing the layout of decision trees inside the ensemble method. Furthermore, the study uses a comprehensive set of measures for assessment of performance, such as TPR, FPR, precision, MCC, recall, and accuracy, to thoroughly assess the presented method’s effectiveness compared to cutting-edge machine learning models like AbM1, J48, KNN, LMT, MLP, NB, and SGD. This presented system differs from the previous research in that it incorporates ensemble learning, GAs optimization, and a rigorous performance evaluation.

Preliminaries

In this section, we provide a brief overview of two foundational techniques: GAs and random forests, that are integral to our proposed model for network intrusion detection.

Genetic algorithms

GAs are adaptive heuristic search algorithms based on the principles of natural selection and genetics. They are particularly effective for optimization problems where the solution space is vast and not explicitly defined. GAs operate by maintaining a population of candidate solutions, which are iteratively evolved to produce better solutions over time. The evolution process in GAs involves several key steps. First, a selection process is carried out, where the best-performing individuals, or solutions, are chosen based on a fitness function that evaluates their quality. Next, a crossover operation is applied, where selected individuals are recombined to create new offspring by combining traits from both parents. This step helps explore new regions of the solution space. Finally, a mutation process introduces random modifications to the offspring to increase diversity within the population, prevent premature convergence, and avoid getting trapped in local optima. Together, these steps ensure that the algorithm effectively navigates the solution space to find optimal or near-optimal solutions.

In the domain of NIDS, GAs are employed to optimize the feature selection process by navigating through the feature space to identify the most relevant attributes. This approach enables the proposed “Opt-Forest” model to construct more compact and accurate decision trees, thereby improving its ability to detect sophisticated and evolving cyber threats. GAs help in balancing exploration and exploitation during feature selection, ensuring that the model does not get stuck in suboptimal solutions and can adapt to diverse attack patterns.

Random forests

Random forests are an ensemble learning technique primarily used for classification and regression tasks. The model builds multiple decision trees during training, each constructed using a random subset of the training data (bagging) and a random subset of features at each split. This randomness introduces diversity among the trees, which reduces the variance and helps prevent overfitting. Each tree in the forest independently votes on the class of a given input, and the final prediction is determined by majority voting (in classification) or averaging (in regression). The strength of Random Forests lies in their ability to handle large datasets with higher dimensionality and their robustness against noise and overfitting.

In the proposed “Opt-Forest” model, random forests serve as the base classifier, with the decision-making process enhanced through optimization techniques like genetic algorithms. By integrating GAs based optimization with the traditional random forest approach, the model achieves greater accuracy and robustness, particularly in detecting subtle and evolving network intrusions. This combination leverages the strengths of both techniques. Random forests ability to generalize across varied data and GAs capacity to optimize feature selection, resulting in a more efficient and reliable intrusion detection system.

Proposed methodology

The rapid evolution of technology has drastically increased connectivity, enabling seamless communication and data exchange. However, this progress has also led to a significant rise in cyber threats. NIDS are essential in this context, as they detect and prevent unauthorized access and malicious activities, ensuring the security and integrity of networked systems. Despite the availability of many attack detection systems, they often fall short in detecting sophisticated and evolving threats, particularly those involving subtle variations or mutations of known attack patterns. To effectively address these challenges, we propose an advanced and intelligent approach to network intrusion detection, designed to enhance detection accuracy and resilience against evolving cyber threats.

System model

To address the limitations of traditional machine learning techniques in network intrusion detection, we introduce an optimized ensemble model for creating decision forests. The primary goal of our study is to offer an ensemble method for network intrusion detection (NID) that uses GAs and decision forests. The proposed system is structured around three core components: Data Gathering and Preliminary Processing, Feature Selection, and Model Training and Evaluation. Figure 1 illustrates the overall system architecture, showing the flow of data and the interactions between components.

Figure 1 Methodology work flow.

Data gathering and preliminary processing: The proposed system begins with the collection of a dataset sourced from the Kaggle repository, which serves as the foundation for the network intrusion detection process. Once the data is gathered, it undergoes a preprocessing step to address any missing values and prepare it for subsequent analysis. This involves cleaning the data to ensure its quality and consistency, followed by organizing it into a format suitable for further processing. The preprocessing phase is critical to ensure that the data is accurate and ready for feature extraction, which directly impacts the effectiveness of the intrusion detection system.

Feature selection: After preprocessing, the system employs advanced feature selection techniques to identify and select the most relevant features from the dataset. This process includes methods such as Best-First Search, Particle Swarm Optimization (PSO), Evolutionary Search, and GS. By integrating these optimization techniques, the system enhances the accuracy, robustness, and efficiency of the decision forest model. The feature selection phase is crucial for improving model performance, as it reduces dimensionality and focuses on the most informative attributes, thereby enhancing the model’s ability to detect network intrusions effectively.

Model training and evaluation: The core of the system involves training the “Opt-Forest” model using 70% of the dataset, while the remaining 30% is reserved for testing the model’s efficacy. The ensemble model integrates decision forests with genetic algorithms to optimize feature selection and enhance detection performance. Following training, the system evaluates the model by comparing it with contemporary machine learning models, including AdaBoostM1 (AbM1), J48, K-nearest neighbor (KNN), logistic model tree (LMT), multilayer perceptron (MLP), naïve Bayes (NB), and stochastic gradient descent (SGD), based on metrics such as true positive rate (TPR), false positive rate (FPR), precision, Matthews correlation coefficient (MCC), recall, and accuracy. The evaluation also includes a comprehensive performance assessment using measures like accuracy, precision, recall, and F-measure. This thorough analysis ensures that the model is robust and effective in identifying network breaches, and its adaptability and reliability in maintaining network security are well validated.

Threat model

The threat model for our proposed “Opt-Forest” model encompasses a broad range of cyber threats, including Denial of Service (DoS) and Distributed Denial of Service (DDoS) attacks, probing and scanning activities. The system is designed to address both external adversaries, who attempt to exploit network vulnerabilities from outside and internal malicious actors, who misuse their legitimate access to execute insider attacks. To counter these evolving threats, the proposed model leverages a combination of ensemble learning and optimization techniques to ensure timely, accurate detection while minimizing false positives and negatives, thus providing robust protection against a diverse spectrum of intrusions.

Proposed “Opt-Forest” model

In this section, we introduce the proposed algorithm, named “Opt-Forest.” This algorithm is specifically designed to optimize an ensemble of decision forests, such as Random Forest, using a GAs framework. This approach leverages genetic algorithms for optimization and decision forests for creating the ensemble model in the context of NID. This combination of techniques aims to improve the accuracy and efficiency of intrusion detection systems by harnessing the strengths of both genetic algorithms and decision forests. The algorithm consists of several phases, each aimed at refining the population of decision trees.

Initialization phase

The initialization phase marks the outset of the optimization process within the “Opt-Forest” algorithm. At this stage, the algorithm lays the groundwork for subsequent iterations by creating three distinct populations: the current population ( Pcur), the temporary current population ( PTempCur), and the modified population ( PMod). Each population serves a unique purpose in shaping the evolution of the decision forest ensemble. Additionally, the algorithm initializes the best-so-far chromosome ( CrSFB), a crucial component tasked with tracking the highest-quality solution discovered throughout the iterative optimization process. By establishing these foundational elements, the initialization phase sets the stage for the iterative refinement and enhancement of the decision forest ensemble to achieve superior performance in network intrusion detection.

Iterative refinement

The algorithm proceeds through J iterations, each consisting of following multiple steps aimed at refining the population of decision trees. Initial population selection: In each iteration, the algorithm performs initial population selection. For odd-indexed chromosomes, stratified sampling is performed to create strata S1, S2, and S3. From these strata, M trees are randomly selected using disproportionate stratified sampling (DSS) to form high-quality chromosomes. For even-indexed chromosomes, M trees are randomly selected from the entire forest, ensuring diversity in the population.

Crossover and Mutation: Next, the algorithm performs “Crossover and Mutation.” Chromosome pairs are selected for crossover, where a 1-point crossover technique generates new offspring chromosomes by combining genetic material from the parent chromosomes. These offspring undergo a 1-bit flipping mutation, introducing variability and aiding the exploration of the solution space.

Elitist Operation: Following this, the “Elitist Operation” step is executed to preserve high-quality solutions. Chromosomes are duplicated into PTempCur, and the best chromosome in the current iteration is stored as CrCurrBest. A crossover is applied again to create PMod, a modified population. The best chromosome in PMod ( CrModBest) is compared with CrSFB; if CrModBest has a better evaluation score, it replaces CrSFB. Additionally, the worst chromosome in PMod is replaced with CrCurrBest if the latter has a better evaluation score, ensuring that elite solutions are carried forward.

Chromosome Selection for the Next Iteration: The algorithm then moves to “Chromosome Selection for the Next Iteration.” A new pool of chromosomes ( PPool) is created by combining Pcur and PMod. From this pool, 20 chromosomes are selected using the roulette wheel selection method, which favors chromosomes with higher fitness scores, ensuring that the next generation starts with a strong set of potential solutions.

Rectification of So Far Best Chromosome: Finally, the Rectification of So Far Best Chromosome (SSO) step is performed. Sequential Search Operations are applied to CrSFB. Each bit in the chromosome is systematically flipped from 1 to 0 or from 0 to 1, checking if the evaluation score improves with each flip. This fine-tuning process helps in identifying and solidifying the best possible solution.

The iterative process continues until J iterations are completed. Throughout the iterations, the combination of stratified sampling, crossover, mutation, elitist operations, and sequential search ensures that the algorithm effectively explores and exploits the solution space, gradually improving the quality of the decision forest ensemble. The final output is a robust ensemble model optimized for detecting network intrusions with high accuracy and low false positives. The Algorithm 1 and Fig. 2 provides a detailed explanation of the proposed “Opt-Forest” algorithm.

Algorithm 1 Opt-Forest algorithm.

 1:  Initialization:	
 2:  Create Pcurr, PTempCurr, and PMod.	
 3:  Initialize CrSFBest.	
 4:  for i=1 to I (Iterations) do	
 5:   Preliminary Population Selection:	
 6:   Selection for odd chromosomes:	
 7:   To create strata St1, St2, St3, do stratified sampling.	
 8:   From strata, randomly select M trees using disparate stratified sampling (DSS) to form CrOdd.	
 9:   Selection for even chromosomes:	
10:   To form CrEven, randomly select M trees from the forest.	
11:   Crossover and Mutation:	
12:   Select pairs (Cri,Crj) of chromosomes for crossover.	
13:   To generate offspring apply 1-point crossover: CrOffspring=Cri⊕Crj.	
14:   Do 1-bit flipping mutation on CrOffspring.	
15:   Discriminatory Operation:	
16:   Double chromosomes to PTempCurr.	
17:   Save the finest chromosome as CrCurrBest.	
18:   To create PMod, apply crossover.	
19:   Compare CrSFBest with CrModBest:	
20:   if EA(CrModBest) > EA(CrSFBest) then	
21:      CrSFBest=CrModBest	
22:   end if	
23:   if EA(CrCurrBest) > EA(CrWorst) then	
24:     CrWorst=CrCurrBest	
25:   end if	
26:   Update CrCurrBest from PMod.	
27:   Select Chromosome for the Next Iteration:	
28:   Create PPool by combining Pcurr and PMod: PPool=Pcurr∪PMod	
29:   Using roulette wheel selection, from PPool select 20 chromosomes.	
30:   Refinement of So Far Best Chromosome:	
31:   Put on Sequential Search Operations:	
32:   for each bit bi in CrSFBest do	
33:    if bi=1 and EA(CrSFBest) improves then	
34:      bi=0	
35:    else if bi=0 and EA(CrSFBest) improves then	
36:      bi=1	
37:    end if	
38:   end for	
39:  end for = 0	

Figure 2 The flowchart of the optimized random forest algorithm.

Data gathering and preliminary processing

The study’s data was obtained from the Kaggle repository, which can be accessed at https://www.kaggle.com/datasets/sampadab17/network-intrusion-detection. The dataset consists of 22,544 rows with 41 columns which initially contains some missing values. To handle these missing values, we applied the mean imputation technique, a common statistical method for handling missing data points.

The mean imputation technique replaces each missing value with the mean (average) value of the observed data within the same feature. Mathematically, this can be represented as:

(1) Meanimputed=∑i=1nxin

where, xi represents the mean values of the features and n represents the observed data points within the features. This equation calculates the average value by summing all observed values and dividing by the number of observations. To illustrate, consider a feature F with missing values xm. For each missing value xm in F: Identify all non-missing values {x1,x2,...,xn} within F.

Compute the mean of these values:

Meanimputed=∑i=1nxin

Replace each xm with Meanimputed.

This method assumes that the data is missing at random, which provides a reasonable approximation for the missing values based on the existing data, thereby reducing the potential bias introduced by missing values and improving the robustness of subsequent analysis and modeling. However, not all of these features significantly contribute to the network intrusion detection process, as some provide limited useful information regarding attacks. To ensure that only the most relevant features are preserved for efficient NID analysis, we performed a careful selection of features utilizing advanced feature extraction approaches, which are covered in the following sections. The selected attributes are presented in Table 1.

Table 1 Selected attributes-a concise display of the chosen features through meticulous feature extraction techniques, optimizing the relevance and efficiency of Network Intrusion Detection (NID) analysis.

Attributes	Attribute position	Description	
Flag	4	Denotes the status or nature of the network connection.	
src_bytes	5	Denotes the quantity of bytes transferred from the origin to the target.	
dst_bytes	6	Represents the amount of bytes that what the target has received.	
logges_in	12	Indicates whether a user is logged into the system (binary value).	
srv_serror_rate	26	Reflects the server’s error rate in servicing connection requests.	
same_srv_rate	29	Specifies the proportion of connections made to the same service.	
dst_host_srv_diff_host_rate	37	Specifies the rate of distinct destination hosts for the same service on the destination host.	

Feature selection

Feature selection is a critical step to improve the performance of the intrusion detection system. We employ a combination of Best-First Search, Particle Swarm Optimization (PSO), Evolutionary Search, and Genetic Search to select the most relevant features from the dataset. The technique of feature extraction involves reducing the dimensionality of a dataset by choosing or manipulating significant variables, typically enhancing model performance and computing efficiency while maintaining vital information for analysis or modelling tasks. This hybrid approach ensures that the selected features maximize the classification accuracy while maintaining the computational efficiency of the model. The selected features are then used to construct the decision forest. The overall methodology for feature selection is presented in Fig. 3. Using each of the previously mentioned searching strategies, a different amount of features is picked. Then the ending process is used, in which the features are chosen by majority vote.

Figure 3 Feature selection methods-depicting the feature extraction process in the study, utilizing four distinct techniques: Best-First, PSO, Evolutionary, and Genetic searches aimed at optimizing model.

Best-first search

Best-First Search (BFS) is a feature selection approach that finds the best subset of features by assessing multiple subsets against a preset criterion, such as the performance of an ML model. To identify the best combination, the BFS algorithm iteratively explores alternative feature subsets, adding or deleting features at each step. During the search process, the aim is to maximize a selected evaluation metric (e.g., accuracy, F1−score). This process is presented in Algorithm 2. BFS enables an exhaustive search over feature subsets, resulting in an optimal or near-optimal feature set for a particular evaluation criterion, improving the efficiency of the NIDS.

Algorithm 2 Feature selection using best first search.

 1:  Set Ebest to an initial value (e.g., 0)	
 2:  while the stopping criterion is not met do	
 3:   for each feature Fi not included in S do	
 4:    Compute the evaluation metric Enew after adding Fi to S	
 5:    if Enew is superior to Ebest then	
 6:     Update Ebest to Enew	
 7:     Include Fi in S	
 8:    end if	
 9:   end for	
10:   for each feature Fi in S do	
11:    Compute the evaluation metric Enew after removing Fi from S	
12:    if Enew is superior to Ebest then	
13:     Update Ebest to Enew	
14:     Exclude Fi from S	
15:    end if	
16:   end for	
17:  end while	
18:  Output the final selected feature subset S = 0	

The given approach combines forward and backward search techniques for feature selection. It investigates the feature space iteratively, assessing how each feature’s addition or deletion affects a selected evaluation measure. Until a predetermined stopping condition is satisfied, this iterative process keeps going. The effectiveness of the algorithm is dependent on the quantity of the feature space and the evaluation metric’s careful selection—which must be inline with the particular goals of feature selection—is crucial to the algorithm’s performance. Although the algorithm provides a methodical way to optimize feature subsets, adding strategies like cross-validation might improve the algorithm’s ability to generalize.

Particle swarm optimization

PSO is an optimization method inspired by nature that is utilized in NIDS feature selection. It uses iteratively updating the location of particles in a multidimensional search space to replicate the social behaviour of birds or fish in order to identify optimal solutions. In the context of feature selection, Every particle denotes a subset of features, and the algorithm looks for the optimal subset that maximizes a specific objective function (e.g., classification, accuracy).

The PSO algorithm, successfully explores feature subsets to identify the best answer based on the provided evaluation measure, making it a useful tool for feature selection in NIDS datasets. The PSO algorithm is decipated in Algorithm 3. In this context, a subset of features, denoted as “subset,” is evaluated by an objective function f(subset) to determine its quality. The function is then employed to return the subset that maximizes the objective function f(subset). The summarize form of PSO is following:

(2) PSO=argmaxsubsetf(subset)

Algorithm 3 PSO search for feature selection.

 1:  Define the PSO parameters	
 2:  num_particles = 100	
 3:  max_iterations = 100	
 4:  weight_of_inertia = 0.7	
 5:  coefficient_of_cognitive = 1.5	
 6:  coefficient_of_social = 1.5	
 7:  Initialize the swarm of particles	
 8:  particles = initialize_particles(num_particles, num_features)	
 9:  Initialize the global optimal position	
10:  global_optimal_position = None	
11:  global_optimal_fitness = −∞	
12:  for range of iterations(maximum_iterations) do	
13:   for particle in particles do	
14:    Update particle position and velocity	
15:    update_particle_position(particle, inertia_weight, cognitive_coefficient, social_coefficient, global_optimal_position)	
16:    Evaluate the fitness of the particle's feature subset	
17:    particle_fitness = evaluate_fitness(particle)	
18:    Update personal optimal position if needed	
19:     if particle_fitness > particle.personal_optimal_fitness then	
20:     particle.personal_optimal_position = particle.position	
21:     particle.personal_optimal_fitness = particle_fitness	
22:     end if	
23:   end for	
24:   Update global optimal position if needed	
25:   if particle_fitness > global_optimal_fitness then	
26:    global_optimal_position = particle.position	
27:    global_optimal_fitness = particle_fitness	
28:   end if	
29:  end for	
30:  The final global optimal position represents the selected feature subset	
31:  selected_features = global_optimal_position = 0	

Algorithm 3 initializes a swarm of particles with predefined parameters, such as the number of particles, the maximum number of repetitions, and the coefficients that govern particle movement. Every particle represents a potential feature subset. It incorporates inertia weight, cognitive coefficient, and social coefficient and iteratively updates the particle placements and velocities depending on their individual and worldwide level optimal positions. Each particle’s fitness is assessed and compared to its own personal highest fitness, which is based on how well its related feature subset performs. The algorithm monitors the global optimal fitness and location for every particle. The final global optimal position, which reflects the chosen feature subset after the predetermined number of iterations, offers a feature selection strategy that optimal strikes an balance between prospecting and extraction in the search space.

Evolutionary search

Another efficient approach for feature selection in NIDS is evolutionary search, which are inspired by biological evolution and entails iteratively evolving candidate solutions (chromosomes) using populations of candidate solutions and genetic operators. The steps of the evolutionary search method for feature selection as shown in Algorithm 4.

Algorithm 4 Evolutionary search for feature selection.

 1:  Define Parameters	
 2:  population_size = 100	
 3:  max_generations = 100	
 4:  breeding_rate = 0.7	
 5:  alteration_rate = 0.1	
 6:  Step 1: Initialization	
 7:  population = initialize_population(population_size, num_features) {Initialize a population of feature subsets}	
 8:  define_evaluation_metric() {Define an evaluation metric}	
 9:  for generation in range(max_generations) do	
10:   Step 2: Evaluation	
11:   evaluate_population(population) {Calculate fitness of each chromosome}	
12:   Step 3: Selection	
13:   selected_parents = select_parents(population) {Select parent chromosomes based on fitness}	
14:   Step 4: Breeding (Recombination)	
15:   offspring = perform_crossover(selected_parents, breeding_rate) {Create offspring from parent pairs}	
16:   Step 5: Alteration	
17:   mutate_offspring(offspring, alteration_rate) {Introduce random changes to some offspring}	
18:   Step 6: Replacement	
19:   population = replace_population(population, offspring) {Replace old population with new population}	
20:  end for	
21:  Step 7: Termination	
22:  {Evaluate the final population}	
23:  evaluate_population(population)	
24:  Step 8: Result	
25:  best_chromosome = select_best_chromosome(population) {Select the best chromosome in the final population}	
26:  selected_features = get_features_from_chromosome(best_chromosome) = 0	

The algorithm starts by creating an evaluation measure and initializing a population of feature subsets. It assesses each chromosome’s fitness in the population repeatedly over a predetermined number of generations. Higher performers are given preference during selection, which is based on fitness. An alteration procedure adds random alterations to certain offspring, whereas breeding is done to specific parent pairings, producing offspring. Until the maximum number of generations is achieved, the old population is replaced by the new offspring population, and so on. The optimal feature subset is represented by the corresponding chosen characteristics. The last stage entails assessing the fitness of the final population and choosing the best chromosome as the solution.

Genetic search

Genetic Search is a strong optimization tool used in NIDS for feature selection. The process of natural selection and evolution inspires. They operate by iteratively evolving a population of candidate feature subsets across numerous generations to identify an ideal or near-ideal subset of characteristics. The method initializes a population of binary chromosomes, uses a GA for feature selection, and assesses the fitness of each population using a predetermined metric. Parents are chosen probabilistically across several generations, and children are produced by breeding and alteration.

In order to optimize feature subsets, the algorithm iteratively replaces the old population with the new one. The optimal feature of a subset that is represented by the top-performing chromosome in the final population is the final output. With the help of this GS technique, the solution space is efficiently explored to find the best feature combination for enhancing model performance. The steps are presented in Algorithm 5.

Algorithm 5 Genetic algorithm search for feature selection.

 1:  Inputs:	
 2:  - size of population (pop_size)	
 3:  - Maximum number of generations (max_generations)	
 4:  - Breeding rate (breeding_rate)	
 5:  - Alteration rate (alteration_rate)	
 6:  Initialize a population of binary chromosomes:	
 7:     population ← Randomly generate 'pop_size' chromosomes (feature subsets)	
 8:  Define a fitness function (fitness) to evaluate the quality of feature subsets:	
 9:    fitness(chromosome) ← Evaluate the chromosome's performance based on an evaluation metric (e.g., accuracy, F1-score)	
10:  Repeat for gen in [1, 2, …, max_generations]:	
11:    Calculate fitness for each chromosome:	
12:     For each chromosome in the population:	
13:       chromosome.fitness ← fitness(chromosome)	
14:    Select parents based on fitness:	
15:     parents ← Select 'pop_size' parents from the population with probabilities based on their fitness (e.g., roulette wheel or tournament selection)	
16:    Generate offspring through breeding (e.g., one-point or two-point):	
17:     offspring ← Perform breeding on pairs of parents with a probability of 'breeding_rate'	
18:    Apply alteration to some offspring:	
19:     Randomly mutate some genes in the offspring with a probability of 'alteration_rate'	
20:    Replace the old population with the new population:	
21:     population ← offspring	
22:  Return the best chromosome from the final population:	
23:     best_chromosome ← Chromosome with the highest fitness in the final population	
24:  Output the feature subset represented by best_chromosome = 0	

Experimental results and analysis

In this section, we present the results of the experiments conducted to evaluate the performance of the proposed “Opt-Forest” model using the UNSW-NB15 dataset. We also provide a comparative analysis of our model against various benchmark models to highlight its effectiveness.

Dataset overview

We utilized the UNSW-NB15 dataset for training and testing our models, which is widely recognized in the field of network intrusion detection. This dataset includes a comprehensive collection of network traffic data that represents both normal activities and various types of intrusions, making it an ideal benchmark for evaluating the performance of intrusion detection systems (IDS). The UNSW-NB15 dataset contains 49 features, including packet-based and flow-based attributes, ensuring a robust and versatile resource for developing and evaluating IDS models. For our study, the dataset was divided into 70% for training and 30% for testing. This split allowed for a substantial amount of data to be used in the training phase, where the model identified patterns and correlations within the data to develop a predictive model. The training subset included both benign traffic and multiple categories of attack traffic, enabling the model to learn the distinguishing characteristics of each. Following training, the model was tested using the testing subset to evaluate its performance and ability to generalize to new, unseen data. In addition to the proposed model, various other models were evaluated as benchmarks to provide a comprehensive evaluation of their effectiveness in achieving the study objectives (Moustafa & Slay, 2015, 2016; Moustafa, Creech & Slay, 2017).

The UNSW-NB15 dataset is critically important due to its comprehensive and diverse collection of network traffic data, which provides a balanced representation of real-world network conditions. Unlike other datasets that may focus solely on specific types of attacks or normal traffic, UNSW-NB15 includes modern attack types such as DoS, fuzzers, analysis, backdoors, exploits, generic, reconnaissance, shellcode, and worms. This diversity ensures that models trained and tested on UNSW-NB15 are well-prepared to handle contemporary security challenges. Additionally, the dataset’s extensive documentation and established benchmarks facilitate reproducibility and comparison across different studies, enhancing its value to the research community. By using the UNSW-NB15 dataset, researchers and practitioners can develop models that are more accurate, reliable, and capable of generalizing to real-world network environments, thereby advancing the state-of-the-art in network security (Moustafa, Slay & Creech, 2017; Sarhan et al., 2021).

Evaluation criteria

In the evaluation of models, the training and testing phases were executed using a 70–30 dataset split. This evaluation encompasses a thorough comparison between the proposed model and a selection of bench-marked machine learning models, which are AbM1 (Subasi & Kremic, 2020), J48 (Maulana & Defriani, 2020; Ortega et al., 2020; Posonia, Vigneshwari & Rani, 2020), KNN (Alroobaea, 2020; Zhang & Li, 2021), LMT (Verma, Yadav & Monia, 2022; Sujal, Nanthini & Reddy, 2022; Bhoyar et al., 2021), MLP (Asiri et al., 2022; Tolstikhin et al., 2021; Yu et al., 2022), NB (Alroobaea, 2020) and lastly, SGD (Toğaçar, Ergen & Cömert, 2020; Stich & Karimireddy, 2020; Upadhyay et al., 2020). The few evaluation criteria are given below:

Precision: The precision of the model is determined by dividing all of its positive predictions by the percentage of true positive rate. It evaluate the model’s accuracy in identifying positive instances without mistakenly labeling negative instances as positive. The precision formula is as follows:

(3) Precision=TPTP+FP

Recall: Recall is a statistic used in classification tasks to assess a model’s capacity to properly identify positive cases from the total number of actual positive examples in the dataset. It is also referred to as sensitivity or true positive rate. Mathematically, recall is calculated as:

(4) Recall=TPTP+FN

F-measure: A statistic known as the F1-score merge accuracy and recall into just one score, hence creating a balance between the two. It is especially effective when the dataset has an unequal class distribution (class imbalance). The formula for the F1-score is:

(5) F−Measure=2∗Precision∗RecallPrecsision+Recall

Matthew’s correlation coefficient: The validity of binary classifications can be gauged using Matthew’s correlation coefficient (MCC). Its range is from −1 to 1. where 0 denotes a prediction that is no better than the random prediction, 1 denotes a perfect prediction, and −1 denotes a complete difference between the prediction and the observation. The formula for MCC is:

(6) MCC=TP∗TN−FP∗FN(TP+FP)(TP+FN)(TN+FP)(TN+FN)

Accuracy: Performance assessment by comparing a predictive model’s predictions to the actual values of the target variable in the dataset, accuracy describes how accurate or how often a prediction is accurate. This statistic is frequently employed to evaluate the overall efficacy of a model in producing accurate predictions. The formula of accuracy is:

(7) Accuracy=TP+TNTP+TN+FP+FN

Results and comparative analysis

To validate the superiority of the “Opt-Forest” model, we compared its performance against several benchmark models, including AbM1, J48, KNN, LMT, MLP, NB, and SGD. In the process of analyzing outcomes obtained from different machine learning algorithms applied to the enhanced dataset containing clearly defined and distinct features, we prioritized identifying the most suitable intrusion detection algorithms, including a novel model. We investigated each of them for finding how much they are accurate to figure out network anomalies and normal activities which are the basic criteria to detecting network intrusions. Models are thoroughly analyzed under critical measurements which are key to ML and which are discussed and analyzed as follows: Firstly, we have calculated the confusion matrix as depicted in Table 2, for each of the algorithms which shows the accuracy of the models in terms of detecting normal activities as “normal” and anomalies as “anomaly” in network intrusion detection.

Table 2 Model performance evaluation-an analysis of employed ML models focusing on their accuracy in distinguishing normal network activities from anomalies, presented through the confusion matrix.

Bold indicates our proposed model.

Model	Class	Normal	Anomaly	
AbM1	Normal	3,690	369	
	Anomaly	132	3,367	
J48	Normal	4,038	21	
	Anomaly	37	3,462	
KNN	Normal	4,000	59	
	Anomaly	38	3,461	
LMT	Normal	4,034	25	
	Anomaly	35	3,464	
LMT	Normal	3,982	77	
	Anomaly	415	3,084	
NB	Normal	3,867	192	
	Anomaly	629	2,870	
SGD	Normal	3,843	216	
	Anomaly	395	3,104	
Opt-Forest (Ours)	Normal	4,046	13	
	Anomaly	29	3,470	

After analyzing the results of different machine learning algorithms, we expanded our research to include crucial metrics required for efficient intrusion detection. We investigated additional performance evaluation like precision, recall, and F1-score in addition to accuracy evaluations. These metrics provide more detailed information on how well the algorithms are able to distinguish between normal and abnormal network activities. Through close analysis of these metrics along with the confusion matrix results, we were able to obtain a thorough understanding of the effectiveness of each method in distinguishing between normal and malicious network activity. The foundation of our methodology for choosing the best intrusion detection algorithms for reliable network security is this kind of thorough assessment.

The algorithms with superior performance are detailed in Table 2. Among them J48 algorithm accurately recognized 4,038 occurrences of the “Normal” class as “Normal,” but incorrectly classed 21 examples as “Anomaly”. It accurately identified 3,462 occurrences as “Anomaly” but incorrectly classed 37 instances as “Normal” for the “Anomaly” class. LMT precisely recognized 4,034 instances of the “Normal” class as “Normal,” but incorrectly classed 25 examples as “Anomaly.” It accurately identified 3,464 occurrences of the “Anomaly” class as “Anomaly,” but erroneously classified 35 cases as “Normal.” The out-performer remained Opt-Forest with the most optimum results. It properly categorized 4,046 occurrences of the “Normal” class as “Normal,” but misclassified 13 instances as “Anomaly.” It accurately recognized 3,470 occurrences of the “Anomaly” class as “Anomaly,” but incorrectly classed 29 instances as “Normal.” The ensemble approach used during data features selection might well be the reason for its better performance.

We extracted results for various machine learning algorithms deployed on a refined dataset with well-distinguished and distinct features. Our focus was on identifying those most suitable for intrusion detection, which included AbM1, J48, KNN, LMT, MLP, NB, SGD, and Opt-Forest model. We investigated each of them for finding how much they are accurate to figure out network anomalies and normal activities which is the basic criteria to detecting network intrusions. Models are thoroughly analyzed under critical measurements which are key to ML and which are discussed and analyzed as: Firstly, we have calculated the confusion matrix as depicted in Table 2, for each of the algorithms which shows the accuracy of the models in terms of detecting normal activities as “normal” and anomalies as “anomaly” in network intrusion detection. The algorithms with better performance are discussed here. Among them, the J48 algorithm accurately recognized 4,038 occurrences of the “Normal” class as “Normal,” but incorrectly classified 21 examples as “Anomaly.” It accurately identified 3,462 occurrences as “Anomaly” but incorrectly classed 37 instances as “Normal” for the “Anomaly” class. LMT precisely recognized 4,034 instances of the “Normal” class as “Normal,” but incorrectly classed 25 examples as “Anomaly.” It accurately identified 3,464 occurrences of the “Anomaly” class as “Anomaly,” but erroneously classified 35 cases as “Normal.” The outperform remained Opt-Forest with the most optimum results. It properly categorized 4,046 occurrences of the “Normal” class as “Normal,” but misclassified 13 instances as “Anomaly.” It accurately recognized 3,470 occurrences of the “Anomaly” class as “Anomaly,” but incorrectly classed 29 instances as “Normal.” The ensemble approach used during data features selection might well be the reason for its better performance. Furthermore, our analysis extends past classification accuracy to consider each algorithm’s robustness and scalability in handling large-scale network datasets. We analyze computing requirements and scalability to ensure that the algorithm chosen is compatible with the organization’s infrastructure and operational requirements.

Performance evaluation is an important phase in machine learning since it allows you to examine the accuracy and efficiency of your model’s predictions. It assists you in determining how well your model is operating and if it satisfies the necessary goals. In this study where, the count of true-positive classification is denoted by TP, while the count of false-negative classification is presented by FN, TN is the count of true-negative classification, and FP is the count of false-positive classifications. In our assessment of each model’s performance, we focused on two crucial metrics, true positive rate (TPR) and false positive rate (FPR), illustrated in Fig. 4, two essential measures in ML, particularly in binary classification issues. These metrics are frequently used to assess the effectiveness of categorization models. TPR indicates how successfully your model distinguishes positive instances from the overall number of positive cases. The fraction of true negative cases that the model mistakenly labels as positive is measured by FPR. Opt-Forest remained an out-performer as it offered 0.994 TPR and a least 0.006 FPR values. This means it classified almost 100% correct occurrences for their true nature and misclassified them for only a fraction of 0.006%.

Figure 4 TPR and FPR analysis-an illustration of TPR and FPR, vital metrics in binary classification models.

Furthermore, the Opt-Forest exceptional performance highlights how well it can distinguish between positive and negative instances, which makes it an effective option in intrusion detection systems. Its remarkably high TPR and quite low FPR figures show how reliable it is at accurately detecting real events while reducing false alarms. Whereas J48 and LMT results are satisfactory, their FPR values were slightly higher than those of the Opt-Forest, suggesting more likely to have for misclassification. This is quite remarkable as compared to other ML approaches evaluated J48 and LMT provided satisfactory results followed by AbM1 and KNN as shown in above Fig. 4.

A separate set of important measurements, including precision, recall, and F-measure, was also extracted for each model on the given dataset and is briefly discussed in the following sections. These metrics serve several functions and provide useful information about the performance of a classification model. Table 3 and Fig. 5 show the critical performance measures, such as precision, recall, and F-measure, for various models utilized in the context of NIDS. Notably, the “Opt-Forest” model consistently outperforms the other algorithms. Its exceptional accuracy (0.994) suggests a low rate of false positives, which is critical in avoiding false alerts for non-intrusive activities. The high recall (0.994) illustrates its capacity to detect real intrusions while reducing the likelihood of missing true threats. Furthermore, the F-Measure (0.994) exhibits a fair choice between precision and recall, confirming the model’s capacity to maintain a harmonic balance between false alarm reduction and intrusion detection. “Opt-Forest” is most certainly benefiting from decision trees’ ensemble nature, utilizing several models to improve accuracy and resilience. This thorough and expert examination highlights the “Opt-Forest” model’s supremacy in NIDS, making it an appealing choice for intrusion detection due to its great overall performance and harmonious balance of precision and recall.

Table 3 Comparison of the proposed model’s performance with various Network Intrusion Detection System (NIDS) models.

Bold indicates our proposed model.

Model	Precision	Recall	F-Measure	
AbM1	0.936	0.934	0.934	
J48	0.992	0.992	0.992	
KNN	0.987	0.987	0.987	
LMT	0.992	0.992	0.992	
MLP	0.938	0.935	0.935	
NB	0.896	0.891	0.891	
SGD	0.920	0.919	0.919	
Opt-Forest (Ours)	0.994	0.994	0.994	

Figure 5 Model performance metrics-A visual presentation of key performance metrics, including precision, recall, and F-measure, for various NIDS models.

Models deployed in this study were investigated for another important ML parameter known as MCC. This is illustrated in Fig. 6. MCC is an acronym for Matthew’s Correlation Coefficient, and It’s a machine learning metric for evaluating how well binary classification models work. Proposed method yielded the most optimum results of 0.989 followed by J48 and LMT with 0.985 and 0.984 respectively. The overall accuracy is a popular indicator for assessing the performance of a classification model. It calculates the percentage of properly identified cases throughout the full dataset. Figure 7 presented the accuracy analysis of each employed model compared with the proposed model. Proposed method outperformed its counterparts in terms of accuracy by achieving 99.45%, followed by J48 and LMT gaining 99.23% and 99.21%, respectively. KNN achieved an overall accuracy of 98.72% followed by AbM1 with 93.37%. Opt-Forest algorithm clearly shows its effectiveness here as this model used an ensemble technique for collecting and analyzing features from the given dataset and this could be the reason behind its top performance.

Figure 6 A visual representation of MCC, a crucial machine learning statistic for evaluating binary classification models employed in this study for NIDS.

Figure 7 A visual representation of model accuracy, highlighting the Opt-Forest better performance.

Percentage difference (PD) is used for extracting model performance comparison (MPC) and finding the algorithms which perform closer to the proposed model. This gives a clue that the models having minimum PD with the proposed model are also suitable and can also be deployed at a similar job. Figure 8 shows PD of the proposed model with all other tested models. PD score of Opt-Forest with J48 is 0.22% and with LMT it is 0.24%. These PD scores give indications that J48 and LMT can also be deployed to attain satisfactory results in the same task. PD scores of other algorithms like NB, SGD, and AbM1 were high and hence cannot be recommended for achieving satisfactory results. Furthermore, PD scores also provide beneficial indicators for finding algorithms that closely match the effectiveness of the proposed method. Via examining PD scores of the various algorithms, researchers can gain insights into which methods offer comparable efficiency in addressing the task at hand. In the context of our study, the relatively low PD scores for J48 and LMT are potential or viable alternatives to the proposed model. However, it’s essential to conduct further analysis to assess the robustness and generalizability of these findings across diverse datasets.

Figure 8 Percentage difference between OPT forest and other employed machine learning models.

Threats to validity

Threats to validity encompass factors that can undermine the accuracy, generalizability, or reliability of research findings. 1. External validity: External validity is critical since it pertains to the generalizability of this research. While the UNSW-NB15 dataset provides a modern foundation, it is important to acknowledge its potential limitations in capturing the full spectrum of network behaviours, as different datasets, network configurations, and industry-specific contexts can result in distinct traffic patterns and intrusion behaviours. As a result, the findings may not apply generally to networks with dramatically different profiles or designs, emphasizing the importance of cautious interpretation and consideration of context-specific differences in network traffic patterns.

2. Internal validity: The integrity of the experimental design and data analysis techniques is important to the internal validity of this study. Particular attention should be paid to any biases or mistakes introduced during data preparation, such as managing missing values and feature selection since these variables might impact the model’s accuracy and validity of the study’s. Furthermore, the effect of parameter and hyper-parameter settings on machine learning model performance, including the “Opt-Forest” model, highlights the importance of explicit documentation of these settings in order to ensure the robustness and replicable of the given findings.

3. Construct validity: Construct validity is concerned with variable measurement and manipulation. In this study, it is crucial to recognize that feature engineering, like choosing and selecting features, have a major influence on the input data for the models. Variations in model performance might result from different feature engineering decisions. The exact collection of characteristics chosen is critical in defining the outcomes and, as a result, the construct validity of the study.

4. Data quality: The accuracy of the data utilized is critical to the research’s credibility. The UNSW-NB15 dataset, albeit more recent in this investigation, may still have limits and potential data quality concerns. Inaccuracies or biases in the dataset have the potential to have a major impact on the study results. Recognizing these constraints and resolving any data quality concerns is critical for maintaining the research’s credibility.

Limitations and future work

The proposed “Opt-Forest” model demonstrates substantial potential for improving network intrusion detection, but several limitations remain. The model’s reliance on high-quality training data means that any biases or gaps in the dataset can adversely impact detection accuracy. Furthermore, the computational complexity of combining multiple optimization techniques, such as genetic algorithms, particle swarm optimization, and evolutionary search, may result in longer training times and require substantial processing power, which could limit its use in resource-constrained environments. Additionally, the model’s effectiveness against entirely novel attack patterns not represented in the training data may be limited, and its performance in real-time scenarios could be affected by network latency and other operational constraints.

Expanding on these limitations, several promising avenues for future research in Network Intrusion Detection Systems (NIDS) emerge. Leveraging advanced machine learning techniques like transfer learning and reinforcement learning could enhance the model’s adaptability to new threats while reducing false positives. Developing larger, more diverse datasets that reflect modern network traffic patterns is also crucial for improving the robustness of intrusion detection algorithms. Additionally, optimizing NIDS deployment in evolving architectures, such as cloud and edge computing, may lead to more effective security solutions. Lastly, future work should prioritize ethical and legal considerations, including privacy and data protection compliance, to ensure responsible NIDS usage. Addressing these areas will enhance the flexibility, efficiency, and ethical integrity of NIDS in safeguarding digital ecosystems.

Conclusion

In this study, we introduced Opt-Forest, an innovative ensemble model designed to bolster NIDS. By integrating genetic algorithms with decision forest approaches and employing advanced feature selection techniques, Opt-Forest overcomes the limitations of traditional machine learning methods in detecting evolving cyber threats. Utilizing the latest UNSW-NB15 dataset underscores the importance of contemporary data in enhancing intrusion detection precision. Opt-Forest demonstrates remarkable effectiveness in balancing precision and recall, minimizing false positives, and achieving near-perfect accuracy in detecting anomalies. Our research highlights the significance of leveraging modern datasets and feature selection methods in developing robust NIDS to fortify cybersecurity systems against dynamic threats. Through comprehensive evaluation against well-known machine learning models, including AbM1, KNN, J48, MLP, SGD, NB, and LMT, consistently outperforms its counterparts. Its integration of genetic algorithms facilitates a broader exploration of solution space, resulting in more accurate and compact decision trees. Advanced feature selection techniques further enhance the model’s robustness, enhancing detection accuracy while reducing false alarms.

Supplemental Information

Supplemental Information 1 Source code and dataset.

Additional Information and Declarations

Competing Interests

Author Contributions

Data Availability

The authors declare that they have no competing interests.

Afaq Ahmed conceived and designed the experiments, performed the experiments, analyzed the data, performed the computation work, prepared figures and/or tables, and approved the final draft.

Muhammad Asim conceived and designed the experiments, analyzed the data, authored or reviewed drafts of the article, and approved the final draft.

Irshad Ullah conceived and designed the experiments, authored or reviewed drafts of the article, and approved the final draft.

Zainulabidin performed the experiments, analyzed the data, prepared figures and/or tables, and approved the final draft.

Abdelhamied A. Ateya analyzed the data, prepared figures and/or tables, and approved the final draft.

The following information was supplied regarding data availability:

The code and models are available in the Supplemental File.

The UNSW-NB15 dataset, a real-world traffic dataset for intrusion detection problems is available at: https://research.unsw.edu.au/projects/unsw-nb15-dataset.

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
