# Peer review of "An optimized ensemble model with advanced feature selection for network intrusion detection"

_PeerJ Computer Science, doi:10.7717/peerj-cs.2472_

## Round 0.1 · original submission · Major Revisions

The paper has merit, though some issues have been pointed out by reviewers. The authors are invited to address the reviewers' comments in their revised version.

Reviewer 1 ·

Basic reporting

In this paper, the authors propose an optimized random forest with GA for network intrusion detection systems. The use of GA is beneficial for mitigating risk by providing more accuracy. They evaluate and prove that the proposed scheme is effective.

Motivation is reasonable and clear in the section of introduction. The contributions are also clear in the section of the introduction.

In the related work section, references with a review of previous works are sufficient. In this section, they present the novelty of the proposed scheme by comparing it with the previous works.

Experimental design

The experimental result clearly verified the effectiveness of the study compared with previous works.

Validity of the findings

The use of a dataset set is robust and controlled. Also, the conclusion is well stated.

Additional comments

Overall, the paper is not bad. Motivation and contributions are clear, and the experimental results are supported. However, system and threat models are missing. To understand the proposed scheme, I suggest adding to the background knowledge, such as GA and random forest.

Reviewer 2 ·

Basic reporting

This paper presents a clear exploration of network intrusion detection, utilizing an innovative ensemble model enhanced by Genetic Algorithms for improved feature selection.
The introduction effectively sets the stage by discussing current challenges in network intrusion detection and outlining the need for more advanced methods. The literature review is comprehensive showing a deep understanding of the subject matter.
The structure of the paper is logically organized, which facilitates understanding and emphasizes the study's contributions.

Experimental design

The research question is pertinent and well-defined, addressing a significant gap in the existing literature by proposing a novel approach to enhance intrusion detection capabilities. The methodology is detailed, employing Genetic Algorithms in a novel way to optimize the decision forest model, which is both innovative and justified.

Validity of the findings

The experimental design is robust and the raw data availability supports the reproducibility of the research findings. The results are presented with clarity. Figures and tables are utilized effectively throughout the paper to aid in explaining the methodology and results.

The paper could benefit from a more detailed discussion of the limitations of the proposed model, showing some of its potential limitations. Expanding on these points could provide a more balanced view and suggest directions for future research.

Additional comments

In addition to addressing the discussed expansions, I recommend that the authors consider citing the following relevant works:

Tama, Bayu Adhi, and Sunghoon Lim. "Ensemble learning for intrusion detection systems: A systematic mapping study and cross-benchmark evaluation." Computer Science Review 39 (2021): 100357.

M. Anisetti, C. A. Ardagna, A. Balestrucci, N. Bena, E. Damiani and C. Y. Yeun, "On the Robustness of Random Forest Against Untargeted Data Poisoning: An Ensemble-Based Approach," in IEEE Transactions on Sustainable Computing, vol. 8, no. 4, pp. 540-554, Oct.-Dec. 2023, doi: 10.1109/TSUSC.2023.3293269.

Reviewer 3 ·

Basic reporting

The paper is well-written and follows a logical structure. The introduction provides adequate background information and context for the research problem. The literature review section covers relevant studies. The methodology section is detailed and explains the proposed approach, dataset, and evaluation metrics clearly. Figures and tables are relevant and support the described methods and results.

Experimental design

The paper is well-written and follows a logical structure. The introduction provides adequate background information and context for the research problem. The literature review section covers relevant studies. The methodology section is detailed and explains the proposed approach, dataset, and evaluation metrics clearly. Figures and tables are relevant and support the described methods and results.

Validity of the findings

The paper is well-written and follows a logical structure. The introduction provides adequate background information and context for the research problem. The literature review section covers relevant studies. The methodology section is detailed and explains the proposed approach, dataset, and evaluation metrics clearly. Figures and tables are relevant and support the described methods and results.

Additional comments

The paper is well-written and follows a logical structure. The introduction provides adequate background information and context for the research problem. The literature review section covers relevant studies. The methodology section is detailed and explains the proposed approach, dataset, and evaluation metrics clearly. Figures and tables are relevant and support the described methods and results.

Reviewer 4 ·

Basic reporting

The paper discusses the challenges of detecting sophisticated cyber threats, where traditional machine learning techniques used in Network Intrusion Detection Systems (NIDS) sometimes struggle. To address these limitations, the study introduces an innovative model called "Optimized Random Forest (Opt-Forest)." This model combines Decision Forest techniques with Genetic Algorithms (GAs) to improve the detection of complex and evolving threats. By employing advanced feature selection methods and leveraging a contemporary dataset, the proposed approach enhances the adaptability and resilience of NIDS. The model's performance is evaluated against several well-known machine learning models, and the results demonstrate its effectiveness and superiority in improving network security.

Even if the work appears to be quite sounding, I have a couple of concerns.


Thus, the first concern the contribution. In particular, since a ton of works about deep-based techniques and feature extraction methods to design intrusion detection systems have been proposed, I suggest to highlight clearly the innovation and the difference w.r.t. the existing literature.

The second one pertains to the experimental analysis. In evaluating the proposed framework the authors refer to classic metrics (e.g. Accuracy, F-Measure, etc.) but a real time complexity analysis lacks.
Specifically, as regards feature selection, such a stage is one hand crucial to make more flexible the prediction, but, on the other hand, such an operation is time consuming itself, whereas some attacks could be performed before the feature selection/extraction task is completed. At this aim, the authors are invited to discuss such a problem and to point some recent literature about this topic. Some suggestions follow:
- "Network Intrusion Detection System using Feature Extraction based on Deep Sparse Autoencoder", (IEEE ICTC conference, 2020);
- "Multi-Stage Optimized Machine Learning Framework for Network Intrusion Detection", (IEEE ,Trans. on Netw. and Serv. Management, 2020);

Minor: enlarge fonts of Fig. 5

Experimental design

The authors employ UNSW-NB15 dataset, to highglith the importance of using modern, real-world data to enhance the precision and effectiveness of intrusion detection systems.

Validity of the findings

The paper should benefit from a time complexity analysis to give more light to the findings.

---

## Round 0.2 · accepted · Accept

Congratulations! Your publication is ready for publication.

Reviewer 2 ·

Basic reporting

no comment

Experimental design

no comment

Validity of the findings

no comment

Additional comments

I have reviewed the revised paper and confirm that all my previous concerns have been fully addressed. The authors have expanded on the model's limitations, providing a more balanced view, and included the suggested references, improving the literature review.
With these revisions, I believe the paper is suitable for publication.

Reviewer 4 ·

Basic reporting

In this revised version, the authors made a nice effort to address my comments raised in the previous stage of review.
In particular, the authors have:
- Clarified the original contributions which basically relies onto Opt-Forest method;
- Clarified the reason behind the choice of not focusing on time complexity analysis.

All in all, i'm satisfied about their response, thus i recommend the acceptance.

Experimental design

N/A

Validity of the findings

N/A